


# Observations of the microphysical evolution of convective clouds in southwest United Kingdom

Robert Jackson[1*], Jeffrey R. French[1], David C. Leon[1,+], David M. Plummer[1], Sonia Lasher-Trapp[2], Alan M. Blyth[3], Alexei Korolev[4]

[1]University of Wyoming Department of Atmospheric Sciences, 1000 E. University Ave, Laramie, WY
[2]University of Illinois at Urbana-Champaign Department of Atmospheric Sciences, 105 S. Gregory St., Urbana, IL
[3]University of Leeds School of Earth and Environment, Leeds, UK
[4]Environment and Climate Change Canada, Cloud Physics and Severe Weather Section, Downsview, ON
[*]Now at: Argonne National Laboratory, Environmental Sciences Division, 9700 Cass Ave, Lemont, IL
[+]Now at: Alpenglow Instruments, Laramie, WY

*Correspondence to:* Robert Jackson (rjackson@anl.gov)

**Abstract.** The COnvective Precipitation Experiment (COPE) was designed to investigate the origins of heavy convective precipitation over the South Western UK, a region that experiences flash flooding due to heavy precipitation from slow-moving convective systems. In this study, the microphysical and dynamical characteristics of developing turrets during four days in July and August, 2013 are analyzed. In situ cloud microphysical measurements from the University of Wyoming King Air and vertically pointing W-band radar measurements from Wyoming Cloud Radar are examined, together with data from the ground-based NXPol radar.

The four days presented here cover a range of environmental conditions in terms of wind shear and instability, resulting in a similarly wide variability in observed ice crystal concentrations, both across days as well as between clouds on individual days. The highest concentration of ice was observed on the days in which there was an active warm rain process supplying precipitation-sized liquid drops. The high ice concentrations observed (> 100 L$^{-1}$) are consistent with the production of secondary ice particles through the Hallett-Mossop process. Turrets that ascended through remnant cloud layers above the 0 °C level had higher ice particle concentrations, suggesting that entrainment of ice particles from older clouds or previous thermals may have acted to aid in the production of secondary ice through the Hallett-Mossop process. Other mechanisms such as the shattering of frozen drops may be more important for producing ice in more isolated clouds.



## 1. Introduction

Raindrop formation at temperatures greater than 0 °C begins with the growth of liquid cloud droplets by condensation and then their further growth to drizzle and eventually rain by collision and coalescence, commonly called the warm rain process. Past observations of ice in warm-based cumulus clouds noted that rapid glaciation often requires the presence of liquid raindrops that can act as embryos for the production of graupel (Koenig, 1963; Hobbs and Rangno, 1990; Huang et al., 2008). These raindrops, once lofted into the updraft above the 0 °C level can freeze, and under the right conditions, generate ice through secondary processes.

Numerous mechanisms have been implicated in the production of ice through secondary processes, defined here as the production of ice through mechanisms not requiring the presence of an ice nucleating particle (Field et al., 2017). In this study we consider two such processes: the first is rime-splintering also known as the Hallett-Mossop process and the second is drop(let) freezing/shattering. The better-known and characterized of these two mechanisms is the Hallett-Mossop process (Hallett and Mossop, 1974). It is active in a narrow temperature regime, -3 to -8 °C, and is believed to require the presence of both actively riming ice particles (typically graupel) and cloud droplets with diameters larger than about 25 µm. Splinter production rates have been quantified based on laboratory measurements allowing this process to be implemented in numerical schemes (Chisnell and Latham, 1976; Mossop, 1978; Cotton et al., 1986; Blyth and Latham, 1997; Huang et al., 2008). Several past studies indicate the importance of rime-splintering for controlling ice number in a range of cloud conditions (e.g. Harris-Hobbs and Cooper, 1987; Blyth and Latham, 1993; Huang et al., 2008; Crosier et al. 2011).

Unlike the Hallett-Mossop process, which requires the presence of graupel and is active over a relatively narrow temperature range, drop freezing/shattering may produce secondary ice particles in regions where graupel is not already present or at temperatures colder than -10 °C. Here we focus on two processes, that, although are different, result from the freezing of a liquid drop. The first is the pure shattering of a liquid cloud droplet or raindrop upon freezing. Early experiments demonstrated that liquid drops with diameters ranging from 30 µm to 1 mm can produce ice fragments through the shattering of an ice shell during freezing and the number of fragments largely depends on the degree of supercooling (Bigg, 1957; Mason and Maybank, 1960). Pruppacher and Schlamp (1975) demonstrated through laboratory experiments that a drop can shatter into various distributions of fragments depending on whether the drop totally or only partially ruptures upon freezing. The process they described can produce "very small" fragments of unknown number and size. More recently, Wildeman et al. (2017) show that raindrops with diameters on the order of 1 mm or larger can explode upon freezing resulting in possibly hundreds of frozen particles. In a related process, Leisner et al. (2014) demonstrated that as drops freeze small 'spicules' could be emitted. In some cases, the freezing of large cloud droplets, with diameters less than 100 µm, could also produce spicules. Both processes have recently been suggested as a potential source of secondary ice in some convective clouds, especially when ice production appears to occur at temperatures too cold for the Hallett-Mossop process to occur (Lawson et al., 2015; 2017). Sullivan et al. (2018) used a parcel model to show that, for clouds with single updrafts and bases warmer than 0 °C and tops as cold as -20 °C, both the Hallett-Mossop process and drop freezing/shattering may be important and that, in general, one does not dominate over the other.



A principal objective of the COnvective Precipitation Experiment (COPE) was to investigate how differences in the strength of the warm rain process impact the development of ice in warm-based convective clouds (Leon et al., 2016). COPE was motivated, in part, by a major flash flooding event that occurred on 16 August 2004, where a line of convective clouds produced peak rainfall rates of more than 300 mm/hr over Boscastle in southwest

England (Golding et al., 2005). Ground-based radar observations of the Boscastle storm suggest tops were likely no higher than -15 to -20 °C, but no in situ microphysical observations were available for this case. Convective clouds that form in SW England typically have bases warm enough (~10 °C) to provide sufficient depth for precipitation to form through collision-coalescence by the time turrets ascend to the level where significant freezing begins (e.g. Huang et al., 2008; 2017). Such conditions are conducive for warm-rain initiation and secondary ice production.

COPE was a two-month field campaign conducted in southwest England during July and August, 2013. Multiple instrumented aircraft collected detailed measurements of cloud microphysical, thermodynamic, and dynamic parameters. A ground-based, scanning X-band radar provided ~5-minute resolution volume scans over the study area (Bennett, 2017). Experiments focused on growing cumulus with cloud tops ranging from 0°C to -25°C.

Three recent studies from COPE are particularly relevant to this work. Plummer et al. (2018) used ground-
based radar measurements to examine the microphysical characteristics of several cases during COPE. Their investigation of the occurrence and structure of ZDR columns implicates that precipitation often formed through collision-coalescence and the resulting large raindrops were then lofted above the 0°C level. Lasher-Trapp et al. (2018) presented results from idealized 3D simulations based on two days from COPE. In their simulations they demonstrated that a stronger warm-rain process produces graupel earlier leading to increased production of ice
through secondary processes. Their simulations showed that the Hallett-Mossop process could produce high ice number concentrations in the modelled clouds, consistent with previous results of Huang et al (2008; 2017), but also that the effectiveness of this process could be inhibited by strong vertical wind shear leading to loss of large particles from the updrafts. Lasher-Trapp et al. (2018) also showed that, while the Hallett-Mossop process could explain the rapid conversion of rainwater to graupel, the amount of precipitation reaching the surface was only minimally
affected by the shift in microphysical pathway from warm rain to graupel. Taylor et al. (2016) examined a single case from COPE focusing on aircraft in situ measurements and found that the first ice particles were frozen drizzle-sized drops and that high concentrations of small ice crystals were subsequently produced through secondary processes.

This study focuses on analyses of the microphysical and dynamical characteristics of four cases where
cloud tops were colder than -10°C and cloud bases were sufficiently warm to allow production of rain through collision-coalescence. These conditions are also expected to be suitable for production of ice through secondary processes. Our examination of observations from these four cases demonstrates substantial variability in the microphysical characteristics of the clouds and allows us to explore the origins of the variability.

In order to describe the microphysical and dynamical properties of growing turrets and compile evidence
for which processes were responsible for hydrometeor growth and development in the observed warm-based convective clouds, we analyze observations from four days during COPE (28 and 29 July and 02 and 03 August,



2013). In Section 2 the data used in this study are introduced and the collection and processing is described. Section 3 presents the main results: first, similarities and differences in bulk cloud properties for the four days are examined; second, using a statistical approach, ice and liquid precipitation particle number concentration and hydrometeor phase are examined to elucidate where and when significant ice production occurred in the clouds; and third,

observations from two individual penetrations are examined in detail to explore how secondary ice production could relate to cloud structure and to examine relationships between the production of ice and other microphysical processes. In Section 4, the results presented in the previous sections are combined to explore the principal mechanisms responsible for producing ice on each of the days. Conclusions are presented in Section 5.

**2. Measurements and Data Processing**

On all four days a trough of low pressure was located west of the study area resulting southerly winds at the surface over the SW peninsula of the UK on one day and southwesterly winds on the other three. Within the southwesterly flow, a sea-breeze that led to an environment favorable for the development of convergence lines formed on 29 July and 02 and 03 August while the southerly flow led to more widespread convective activity on 28

July (Fig. 1). The clouds on these days had similar cloud base temperatures (from 9 to 12 °C) and tops as cold as -25 °C. The data used in this study come primarily from measurements collected onboard the University of Wyoming King Air (UWKA) research aircraft as it made penetrations in clouds at and above the 0 °C level (University of Wyoming Research Flight Center; 2016a,b). Leon et al. (2016) provide a comprehensive list of instrumentation carried onboard the UWKA as part of COPE. Here we discuss only on those measurements directly relevant to our

analysis.

**2a. Airborne and ground radar data**

Data from two ground-based radars provided information on cloud structure. The National Centre for Atmospheric Science (NCAS) ground-based X-band radar (NXPol), located near Davidstow in the southwest United

Kingdom as shown in Figure 2 of Leon et al. (2016), was used to provide general information about cloud structure and stage of development (Bennett, 2017). During the four days analyzed in this study, the NCAS radar performed plan position indicator (PPI) scans at 10 elevations ranging from 0.5° to 9.5° at 1° intervals. The scans covered an area of approximately 100 km by 100 km at a typical spatial resolution of 200 m. Approximately five minutes was required to complete a single volume scan. The Python ARM Radar Toolkit was used to visualize the X-band radar

data (Helmus and Collis, 2016).

The Wyoming Cloud Radar (WCR; Wang et al., 2012; University of Wyoming Research Flight Center, 2016b), an airborne W-band radar installed on the UWKA with two near-vertical beams (up and down), measured radar reflectivity and the near-vertical component of Doppler velocity. Profiles were provided roughly 15 times per second, approximately every 6 m along the UWKA flight track at nominal research flight speeds (90 m s$^{-1}$).

Processing of data from the WCR included thresholding all data at 3 standard deviations above the noise and



removal of surface returns and ground clutter. Doppler velocities were corrected for the motion of the aircraft (Haimov and Rodi, 2013).

Echo-top height was estimated from WCR measurements following the methodology of Rosenow et al. (2014) and Plummer et al. (2015) using the texture σ of the Doppler velocity of the 8 adjacent pixels surrounding each point. Reflectivity factor from W-band radars is strongly affected by attenuation from cloud and drizzle droplets (Lhermitte, 1990; Pujol et al., 2007) and even more severely from raindrops (Lhermitte, 1990). No attempt was made to correct for attenuation, as these data are used qualitatively to describe cloud structure near the aircraft.

*2b. UWKA in situ measurements*

A Droplet Measurements Technology (DMT) Cloud Droplet Probe (CDP) sampled particles with diameter, $D$, $2 < D < 50$ μm and derived their sizes from the intensity of forward scattered light assuming spherical water droplets and Mie-Lorenz theory (Lance et al., 2010). Cloud liquid water content (LWC) was derived from the third moment of the size distribution measured by the CDP. Comparisons between CDP-derived LWC and that from various bulk methods including a DMT LWC-100 hotwire, a Nevzorov probe (Korolev et al., 1998), and a Gerber Particle Volume Monitor-100A (PVM; Gerber 1994) show agreement generally within 10 to 15 % over the entire COPE campaign (Sulskis et al., 2016; Faber et al., 2018).

Two optical array probes (OAPs) were used to derive information about hydrometeors larger than a few tens of microns. A gray-scale Cloud Imaging Probe (CIP-Gray), with tips designed to mitigate contamination by shattering of ice on the probe tips (Korolev et al., 2013), captured two-dimensional gray-scale silhouettes of particles with a nominal range of $25 < D < 1600$ μm. A 2D Precipitation (2DP) Probe captured two dimensional images of particles with a nominal range of $200 < D < 6400$ μm.

A Reverse-flow temperature probe provided a measure of temperature. Vertical wind was derived from measures provided by a 9-hole gust probe and a coupled GPS-INS inertial reference system (Leon et al. 2016, online supplement).

## 2b. OAP processing strategy

Data acquired by the CIP and 2DP were processed using the University of Illinois OAP Processing Software (UIOPS) described in detail by Jackson et al. (2014). Although the CIP installed on the UWKA recorded shaded intensity at 3 threshold levels for each pixel, we consider only the 50% threshold level rendering the data the same as that provided by a standard binary OAP. Due to the poorly defined depth of field of OAPs for small particles (Baumgardner and Korolev; 1997), concentrations of particles of $25 < D < 100$ μm from the CIP are not reported in this study.

Korolev et al. (2011) demonstrate the impact on OAP-measured size distributions due to particle shattering on probe tips and inlets. Korolev et al. (2013) and Jackson et al. (2014) estimated that shattering can cause particle number concentrations to be overestimated by up to an order of magnitude and recommended that mitigation approaches should include both modified probe tips coupled with processing algorithms to identify and remove shattered artifacts. The CIP was equipped with modified 'anti-shatter' tips and the inter-arrival time algorithm of





Field et al. (2006) was applied during processing of the OAP data. In order to determine the threshold used to identify shattered artifacts, an analysis of inter-arrival times was first applied to time periods where only ice was seen in the CIP imagery and the CDP number concentrations were < 1 cm$^{-3}$ in order to ensure that liquid particles were not present. Based on this analysis, all particles with inter-arrival times of less than 10$^{-5}$ s were rejected as

artifacts. These contributed about 7% to the total number concentration.

To distinguish between spherical particles (assumed to be liquid) and ice particles, the habit identification algorithm of Holroyd (1987) was used and applied to CIP images. This algorithm sorts particles into nine categories: sphere, tiny, linear, oriented, graupel, aggregate, irregular, and hexagonal. However, in order to reduce the possibility of misidentification of ice due to over-categorization, in this study, particles are classified as either spheres (likely

liquid) or non-spherical (ice). Images with areas of less than 100 pixels that are classified as tiny by the Holroyd algorithm are not included in the spherical/ice categorization for this study. For the CIP this corresponds to hydrometeors with diameters less than roughly 250 μm. We further restrict this analysis to particles that are entirely imaged within the diode array, to reduce misclassification due to partially-imaged particles. The spherical classification may contain some ice, especially recently frozen drops and lightly rimed graupel; however the non-

spherical (ice) classifications will rarely contain significant concentrations of liquid hydrometeors. Habit identification was not applied to 2DP data because graupel and spherical raindrops are nearly impossible to distinguish due the coarse resolution of the probe.

### 3. Results

Here, we present analyses of measurements obtained in convective clouds sampled on four days during COPE (28, 29 July and 02, 03 August). These days were chosen because the clouds grew above the 0 °C level, providing an environment where ice formation is possible. Despite broadly similar synoptic and thermodynamic conditions on these days, the microphysical evolution of the clouds differed significantly from one day to another particularly with respect to ice formation.

The observations will show that, on each day, as clouds ascended up to and above the 0 °C level, the initial development of precipitation resulted from the warm rain process: condensation followed by collision-coalescence, in agreement with conclusions based on analysis of ZDR columns, corresponding to regions of millimeter-sized drops, presented by Plummer et al. (2018). However, concentrations of millimeter-sized drops were significantly less on 02 August than on the other three days. Measurements at or just above the 0 °C level reveal the presence of

mm-diameter drops on 28 and 29 July and 03 August, but not on 02 August until the clouds ascended to about -6 °C; even then drop concentrations were orders of magnitude less than on the other three days.

In all cases, ice particles began to appear in measurable concentrations by the time cloud tops reached -6 to -8 °C. Here too, significant differences between the four days were observed. On 28 July, significant ice production occurred at higher temperatures as the precipitation (defined as particles with D > 300 μm) was composed almost

entirely of ice by the time cloud tops reached -10 °C. On the other end of the spectrum, most of the precipitation in clouds sampled on 02 August remained liquid at T = -13 °C, the coldest level sampled on that day. Observations



from clouds on 29 July and 03 August fall between these two extremes. On 03 August, more than half of the clouds sampled were composed mostly of ice precipitation as clouds tops cooled to -8 to -10 °C. In contrast, on 29 July, much less than half of the precipitation particles appeared to be ice at the -8 °C level, and glaciation occurred more slowly with height as roughly half of the clouds sampled at -12 °C were still dominated by liquid precipitation at that level.

In the following sections, we investigate environmental factors that may be responsible for the observed differences in ice and liquid precipitation development on the four days.

### 3a. Overview of cases and environmental conditions

The four days in this study had similar aircraft-observed cloud base temperatures, ranging from 9 to 12 °C, with July 28 being colder than the rest of the days (Table 1). On each of the days, the UWKA conducted penetrations of growing cumulus clouds within 60 km of the NCAS radar between the 0 and -15 °C level. The tops of rising turrets were penetrated as they first began to reach the level of the UWKA. Repeated sampling through the same cloud was generally avoided. Growing turrets were targeted visually from the cockpit with the intent of penetrating clouds within 1 km of cloud top. For all of the penetrations analyzed here, 81% of the WCR-estimated echo-top heights are within 1 km of the UWKA flight level; some examples are shown in Fig. 2.

Individual penetrations are defined by periods with $LWC > 0.05$ g m$^{-3}$ for at least 300 m and separated by more than 100 m. Time periods with $LWC > 0.05$ g m$^{-3}$ separated by less than 100 m are classified as one penetration. The total number of penetrations was 225, with the number of penetrations ranging from 47 on 28 July to 66 on 02 August (Table 1).

In order to investigate cloud microphysical conditions in ascending regions of cloud, some of the following analyses were restricted to measurements obtained in updrafts. For this, an updraft is defined as a region within a penetration where the vertical velocity exceeds 1 m s$^{-1}$ over a continuous region at least 300 m wide. Further, the maximum updraft within that same region must exceed 3 m s$^{-1}$. Using these criteria, a total of 84 updrafts were identified on the four days, ranging from 13 on 03 August to 34 on 02 August (Table 1)[1]. More restrictive criteria (larger updraft speeds) would have significantly reduced the number of updrafts identified on all of the days except 02 August.

Although the median value of the maximum updraft speed was greater on 29 July compared to 02 August, the percentage of penetrations containing updrafts was significantly greater on 02 August. The median maximum updraft and the percentage of clouds with updrafts were less on 28 July and 03 August. This corresponds to differences in CAPE (Table 1) that led to significant differences in observed cloud depths on the four days, with 29 July and 02 Aug having strongest updrafts and highest (coldest) observed cloud tops. However, Table 1 also demonstrates significant variability in updraft velocities on a given day.

---

[1] Not all penetrations contain updrafts meeting our criteria, hence 'updrafts' make up only a subset of 'penetrations.'





There was large variation in the environmental shear and CAPE through the cloud depth for the four days. In all cases, there was little turning of the wind with height and thus the shear was aligned with the mean wind direction. The convective lines that formed on 29 July and 02 and 03 August were aligned with the wind direction. This provided an environment favorable for the formation of linear convective systems such as those shown in Fig. 1 for these three days.

The four cases considered here represent a spectrum of cloud strength. Clouds that formed on 28 July were the weakest, growing in environments with significantly less CAPE and vertical wind shear than the other three days as seen in Table 1. This resulted in clouds that were both shallower with weaker updraft velocities and which tilted less with height. The nature of the convection itself was also different compared to the other three days, being more widespread and less organized. Weaker shear may enhance precipitation growth, as precipitation that forms within the updraft can later fall back through the updraft collecting additional cloud liquid water. Conversely, greater shear may cause precipitation to fall outside of the cloud resulting in conditions less conducive for secondary ice production and growth. Indeed, numerical modeling of the 02 and 03 August cases (Lasher-Trapp et al., 2018) suggested that the strong vertical wind shear on 02 August would be less favorable for secondary ice production than the much weaker shear on 03 August.

At the other end of the spectrum, both CAPE and vertical wind shear were significantly greater on 02 August than on the other days in this study. Similar to 29 July and 03 August, clouds formed along a line, but on 02 August, the line was relatively narrow (Fig. 1) resulting in clouds that were more isolated[2], and leaned much more with height. Clouds on 29 July and 03 August reside between these two extremes. On both days, clouds grew in an environment that was only weakly sheared, and clouds grew within lines that were more filled-in containing clouds more closely spaced together.

Resulting differences in the detailed microphysical structure based on in situ observations from these four study days are analyzed in the next section. In particular, we consider how the differences in the cloud dynamics between the cases described above may explain the evolution of the precipitation and the productivity of warm rain and ice processes.

### 3b. Cloud Microphysical Characteristics

For each of the 225 cloud penetrations on the four days (a subset of which contained 84 updraft regions), statistics related to hydrometeor concentration and particle shape were computed and stratified by day and temperature. These results are shown in Figs. 3 and 4. There does not appear to be any systematic difference in the microphysical characteristics between observations obtained from cloud penetrations without updrafts (closed circles) and those only from updrafts (open circles). Therefore for the following discussion we consider observations from all 225 cloud penetrations.

---

[2] The clouds were more isolated at the time of the UWKA flight. Later in the day on 02 August, aftr the UWKA had landed, the line filled-in and this may have impacted precipitation processes as the cells became more closely packed along the line.



For penetrations between 0 and -3 °C, median concentrations of hydrometeors with D > 100 μm (D > 300 μm) were roughly one to two orders of magnitude greater on 28 and 29 July and 03 August than on 02 August (Fig. 3). On 28 and 29 July, when there were enough identifiable particles present at this level to identify habit, less than 20% of the particles were aspherical, indicating they were likely liquid drops (Fig. 5). CIP imagery also clearly

indicates the presence of raindrops with diameters exceeding 1 mm on these two days. No such large raindrops were sampled at this level on 02 August suggesting that there was a less active warm rain process in the growing turrets on 02 August compared to the other days. In fact, although several penetrations were made at this level on 02 August, none contained sufficient numbers of particles to allow a statistical computation of particle shape (hence no data from between 0 and -3 °C in 02 August are shown in Fig. 3). Only a few penetrations were made by the UWKA

at this level on 03 August. However, Taylor et al. (2016) reported up to 50 L$^{-1}$ of spherical precipitation near 0 °C from the BAe-146, providing evidence that the raindrops were being produced through collision-coalescence on this day.

In the Hallett-Mossop zone (Hallett and Mossop, 1974), between the -3 and -8 °C level, concentration of precipitation-sized particles increases relative to the 0 to -3°C level for all four days. For hydrometeors with D > 100

μm (Fig. 3, blue dots), concentrations were roughly the same at -8 °C on all of the days. However, for larger hydrometeors, with D > 300 μm (Fig. 3, red dots), concentrations on 02 August remain one to two orders of magnitude less than on the other three days.

Significant differences are also found in the percentage of particles imaged by the CIP that were identified as aspherical (Fig. 4). In most penetrations on 28 July, the majority of particles sampled near -8 °C were aspherical

(ice). On 03 August, CIP observations from most of the penetrations at this level also revealed that the majority of particles were ice, although a few penetrations contained only a small percentage of ice hydrometeors. For those penetrations that did contain ice, images from the CIP suggest a mix of graupel and small columnar or linear-type crystals similar to that reported by Taylor et al. (2016) based on measurements from 03 August only. On both 29 July and 02 Aug, generally less than 20% of the hydrometeors were aspherical, indicating that most of the precipitation

remained in liquid form at this level. Images from the CIP do indicate the presence of a few small columnar-shaped crystals at -8 °C on 29 July too small (D < 300 μm) to be considered in the shape analysis, but suggesting some ice is present here.

Just above the Hallett-Mossop zone, between -8 and -10 °C, nearly all of the penetrations on 28 July were dominated by ice precipitation. At even higher levels (-10 °C level and above) the majority of penetrations on 03

August (approximately 75%) had precipitation dominated by ice. Contrary to this, penetrations on 29 July and 02 August remained largely dominated by liquid precipitation. It was not until temperatures reached -12 °C, well outside of the Hallett-Mossop zone, that we began to see many penetrations dominated by ice on 29 July. Penetrations on 02 August remained largely devoid of ice even at temperatures less than -12 °C, suggesting that the processes responsible for ice production on 28 July, 29 July, and 03 August were likely less active on 02 August.

Figure 5 shows representative hydrometeor size distributions from updraft cores at several temperature levels for each day. For penetrations near the -3 °C level (red lines), the concentration of cloud droplets with




diameters greater than 30 µm is an order of magnitude less on 02 August than on any of the three other days. This is likely a due to the larger cloud droplet number concentration observed at cloud base on 02 August (Table 1) and is consistent with a slower collision-coalescence process as expected given the smaller median droplet diameter, narrower droplet spectra, and stronger updraft velocities. Hence, a greater cloud depth was necessary to grow

precipitation-sized drops on 02 August compared to the other days, explaining the lower concentrations of 300 µm and larger particles seen on 02 August in Fig. 3.

On all days, the concentration of particles with diameters greater than a few hundred microns increased significantly at the -6 to -8 °C levels (green lines) compared to their concentrations lower in the cloud. However, on the days when more ice was present at these levels (28 July and 03 August) the increase in concentration, by more

than an order of magnitude, was significantly greater than on the days when little or no ice was present (29 July and 02 August).  The presence of particles up to a few millimeters in diameter at -6 to -8 °C further supports the idea that graupel and frozen drops likely provided the rimers necessary for secondary ice production. At the -10 °C level (blue lines), hydrometeor size distributions extended to a few mm on 29 July and 03 August, with lower particle concentrations and mostly liquid precipitation on 02 August. Thus, the lack of millimeter-size hydrometeors could

have inhibited secondary ice production on 02 August.

The picture that emerges from this analysis is that formation of precipitation sized drops through collision-coalescence low in the cloud appeared to be less efficient on 02 August resulting in fewer drizzle and rain drops by the 0 °C level than on the other three study days. This further retarded growth of larger hydrometeors higher in cloud and ice production through secondary processes. Precipitation sampled by the UWKA remained mostly liquid for

the clouds sampled on 02 August even down to temperatures near -14°C. Precipitation formation proceeded rapidly to the ice phase on 28 July and 03 August at temperatures consistent with secondary production through the Hallett-Mossop process. The production of significant ice took place at lower temperatures on 29 July, outside of the Hallett-Mossop zone, suggesting a different process such as drop freezing/shattering might produce the high ice concentrations observed.

### 3c. Possible Ice Enhancement by Recirculation

The observations demonstrate a large degree of variability not only between days, but also from a single day at a given level. To further investigate the factors impacting ice formation in the COPE clouds, we closely examine observations from two individual penetrations at the -8 °C level. These penetrations, from 29 July, were

chosen because the large range of ice crystal concentrations observed in the two clouds together roughly covers the total range measured in all of the COPE clouds from the four days at -8 °C. Further, the two clouds show differences in the predominant precipitation phase. In the penetration shown in Fig. 6, precipitation is all-ice with a mix of graupel and columnar-shaped crystals. For the penetration shown in Fig. 7, precipitation appears to be mostly liquid ranging in size from a ~300 µm to 1 mm in diameter with a few graupel particles.

PPI scans from the NCAS radar indicate both clouds were relatively isolated from larger complexes located to the north and east and near-surface reflectivity values were approaching 50 dBZ during the two



penetrations (Figs. 6c and 7c). Vertical cross-sections from the WCR indicate for both penetrations, the UWKA passed roughly 0.5 to 1 km below cloud top. The region devoid of radar echo below the cloud (Figs. 6, 7a and b) is due to strong attenuation that is expected in presence of precipitation-sized liquid drops at W-band (Lhermitte 1990).

5        The WCR images also show that echoes surrounding the updrafts were present at altitudes up to 4 km in the ice phase cloud (Fig. 6a and b), but only extend up to 3 km in the cloud composed of mostly of liquid precipitation (Fig. 7a and b). The 0 °C level is roughly 3.2 km in both clouds. In both cases, the updrafts sampled by the UWKA would have grown through the cloud layer seen by the WCR. Any particles entrained into the updraft in Fig. 7 would likely be liquid, since the surrounding cloud existed entirely below the freezing level. However, if the higher growth for the surrounding clouds in Fig. 6 contained ice, then these ice particles could be entrained into the
growing updraft and immediately begin to interact with the raindrops already present.

       Cross-sections constructed from the NCAS radar scans (Fig. 8, left column) show that the cloud in Fig. 6 grew in the vicinity of another cloud that had echo top heights of around 5 km about 10 minutes before the UWKA penetration at 1237 UTC. Meanwhile, the cloud in Fig. 7 grew in a region of radar echoes extending up to about 3 – 3.5 km (Fig. 8, right column) and it was not until just a few minutes before the UWKA penetrated the cloud at 1242
UTC that any echoes higher than 4 km were detected. This type of growth was relatively common in COPE clouds where turrets, while relatively isolated at the level of the UWKA penetrations, developed in clusters such that the turrets that ascended to the UWKA flight level often passed through regions of existing cloud.

       The two penetrations considered here contained similar amounts of cloud liquid water within their updraft cores, roughly 1 to 2 g m$^{-3}$ (Figs. 6e, 7e). Both clouds had peak updrafts between 10 and 15 m s$^{-1}$ (Figs. 6d and 7d
respectively). The stronger 15 m s$^{-1}$ updraft in Fig. 7d is consistent with more vigorous growth of the cloud suggested by the NCAS radar imagery. However, the concentration of hydrometeors with D > 100 μm ranged from a low of 50-100 L$^{-1}$ in the cloud composed of mostly liquid precipitation (Fig. 7d) to in excess of 300 L$^{-1}$ in the cloud containing all ice-phase precipitation particles (Fig. 6d). CIP images (Figs. 6f) show the presence of graupel and columns in this cloud, consistent with what would be expected if secondary ice production and riming were
occurring (Hallett and Mossop, 1974). In fact, 94% of the identifiable particles in the updraft were identified as ice. However, in the cloud containing mostly liquid precipitation, only 33% were identified as ice, and what few ice particles were detected were mostly large graupel (Fig. 7f).

       Recycling of ice into the cloud shown in Fig. 6 would increase the amount of ice available for riming and secondary ice production through the Hallett-Mossop process. On the other hand, for the case in which ice recycling
was unlikely to occur (based on the WCR and NCAS images), secondary ice production could only proceed after ice was introduced through primary nucleation. In Fig. 7, we note the presence of large frozen graupel and a few smaller ice crystals, but most of the hydrometeors (regardless of size) appear to be spherical and likely remain liquid at the -8 °C level of the penetration within the strong updraft.

**4. Implications for ice production in COPE clouds**



Observations of ice crystal concentrations range from hundreds per liter near cloud top at -8 to -10 °C on 28 July and 03 August to tens per liter on 29 July. Such values are up to four orders of magnitude greater than the ~0.1 $L^{-1}$ predicted by the ice nuclei parameterization of DeMott et al. (2010) at -10 °C. Such high concentrations are consistent with observations reported in other studies in convective clouds over SW England (Taylor et al., 2016;

Huang et al., 2008, 2017). Ice production through secondary processes is likely controlling the amount of ice in these clouds. However, differences in where and when ice was produced and grew to precipitation-sized particles on the three COPE days suggests that the dominant mechanisms of ice production may differ from day to day and even between clouds on any given day.

Of the four days considered in this study, glaciation proceeded most quickly on 28 July. Clouds on this day

were the shallowest and least vigorous of the four cases and formed in an environment of widespread, unorganized convection devoid of shear. Precipitation developed through collision-coalescence by the time cloud tops reached -3 °C, raindrops larger than 1 mm in diameter were found throughout the clouds at this level. Nearly all clouds sampled on this day were dominated by frozen precipitation by the time their tops reached the -8 to -10 °C level. The presence of millimeter-sized raindrops prior to the production of significant concentrations of ice-phase precipitation

that occurred in the Hallett-Mossop zone is consistent with observations from several recent studies (Taylor et al., 2016; Lasher-Trapp et al., 2016; Heymsfield and Willis, 2014; Huang et al., 2017) that implicate the Hallett-Mossop process as the principal cause of glaciation in some convective clouds. The other days from COPE that show some amount of glaciation in cloud tops between -8 and -10 °C (29 July and 03 August), also produced millimeter-sized raindrops at the -3 °C level. However, glaciation proceeded somewhat differently on these two days, compared to 28

July.

The Hallet-Mossop process requires the presence of some initial ice particles to initiate the multiplication process. The source of this initial ice is often believed to be primary nucleation of the raindrops as discussed in the previous paragraph. However, environments containing detrained ice or wide-spread cloud that extend above the environmental 0 °C level also provide a source for initial ice. Idealized modeling studies of the 3 Aug case by Moser

and Lasher-Trapp (2017) suggest that the cloud forcing and weak vertical wind shear would allow for both possibilities. The conditions on 28 July were conducive for entrainment of ice particles from outside of the sample turret (see for example Fig. 2a). On this day, ascending turrets often rose through regions in which older clouds were present. Because the turrets were ascending in a low-shear environment, these newly frozen raindrops would remain in the center of the updraft, collocated with high cloud water content. Observations from 29 July and 03 August are

also consistent with this idea. However, because the clouds on these two days formed along lines, some clouds sampled on these days were more isolated than others. Figs. 6 and 7 provided two examples from 29 July in which turrets ascended through the remnants of other clouds. In one of these cases, the older cloud remnants extended above the environmental 0 °C level and likely contained ice, the precipitation sized particles were frozen by the time the top ascended to -8°C. Taylor et al. (2016) concluded that for clouds sampled along the convective line on 03

August, those clouds on the downwind end of the line were more glaciated than new growth that was forming on the upwind end of the line. All of this suggests, that at least for the clouds in this study, the presence of a source of initial



ice helped jump start the Hallett-Mossop process and rapid freezing of precipitation sized particles observed in some of the clouds.

Most of the clouds sampled on 29 July and some of the clouds sampled on 03 August did not exhibit much ice production by the -10 to -12 °C level. Most of the precipitation in these clouds did not freeze until cloud tops had

reached between -12 and -14 °C, suggesting that a mechanism other than the Hallett-Mossop ice multiplication process was likely responsible for ice production. In these cases, turrets were more isolated and remnants from earlier clouds did not extend significantly above the 0 C level. The observation of rapid transition to ice-phase precipitation occurring at -12 °C is more consistent with observations reported by Lawson et al. (2015; 2017) who interpreted their observations as the result of drop freezing/shattering. It is possible that drop freezing/shattering

could also take place at lower levels in cloud, but one might expect that the large droplets freeze more easily at lower temperatures, which could explain why we do not observe transition to ice-phase in these cases until -12 °C.

Very few of the clouds sampled on 02 August contained any significant concentrations of ice, even as tops approached the -14 °C level. A significant difference on this day was the lack of precipitation-sized liquid drops. However, the detailed in situ observations reported here are somewhat at odds with those reported by Plummer et al.

(2018) who showed ZDR columns with values up to 3 dB extending up to 1 km above the 0 °C level on this day. Such high values suggest raindrops were present at least in some clouds on 02 August. Our analysis does not completely preclude the existence of raindrops on this day, but rather suggests that the concentration of such drops was significantly less than on the other three days examined in this study.

The locations of raindrops on 02 August in relation to updrafts may have influenced ice production. Clouds

were most isolated on this day compared to the other three days, thus hydrometeor recycling was not likely. Further, because of relatively strong shear that was present on 02 August, any precipitation that did form would be less likely to remain within regions of updraft and cloud liquid water. Lasher-Trapp et al. (2018), created idealized simulations representative of 02 and 03 August, and reported that precipitation that was produced in the 02 August clouds was transported downshear and fell mostly outside of the cloud. This suggests that the in situ observations discussed may

under-report the amount of precipitation produced on 02 August. It may also be, that the strong shear present on 02 August, leading to strongly tilted cloud turrets, resulted in an aircraft sampling strategy that favored 'upwind', more isolated clouds. Finally, the UWKA left the area before the convergence line on 02 August produced more pronounced precipitation, which also can contribute to the differences in the occurrence of large drops observed by Plummer et al. (2018) and this study.


## 5. Summary and Conclusions

The analyses presented focus on measurements within developing convection in Southwest England from four days in July and August, 2013.  The UWKA, equipped with *in situ* particle measurement probes and a profiling W-band cloud radar, penetrated turrets within a few hundred meters of their tops as they ascended through the 0 to -

15 °C level. Measurements from individual penetrations provide snapshots in time and, taken together, provide details about the cloud microphysical properties in the developing storms. Large variability in the microphysical



parameters was observed on these four days, indicating significant differences in processes responsible for ice production in these clouds.

1. The greatest amount of ice was observed on the days in which there appeared to be a vigorous warm rain process, based on measurements made at temperatures exceeding -3 °C. This is consistent with past studies suggesting that the production of rain through collision-coalesence is crucial for providing graupel embryos required for secondary ice production to occur in developing turrets.

2. The high (greater than 100 L$^{-1}$) ice concentrations observed between the -8 and -10 °C level in nearly all clouds on 28 July, most clouds on 03 August, and only a few clouds on 29 July are consistent with the production of ice through the Hallett-Mossop process.

3. In addition to a strong warm rain process, the Hallett-Mossop process appeared to be aided by turrets that ascended through regions of cloudy remnants extending above the 0 °C level. Such regions could provide, through entrainment, a necessary source of ice crystals for initiating the Hallett-Mossop process.

4. In cases that did contain raindrops but turrets were more isolated, the ice concentrations measured at the -8 to -10 °C level suggest that the Hallett-Mossop process was less effective at producing ice precipitation. In such cases, glaciation did occur, but not until turrets reached -12 to -14 °C. In such cases, secondary production may occur through another mechanism such as drop freezing/shattering.

5. Clouds on 02 August were much less efficient at producing precipitation through warm-rain, presumably due to higher droplet concentrations. This, in turn, reduced efficiency in producing ice (both through the Hallet-Mossop process and through drop freezing/shattering) due to the much lower raindrop concentrations.

The conclusions based on this work are in general agreement with others from previous studies (e.g. Blyth and Latham, 1997) and also from observation and modeling studies based on the COPE data set that showed the importance of secondary ice production (Taylor et al., 2016; Lasher-Trapp et al., 2018), the influence of cloud spacing and potential hydrometeor recycling (Moser and Lasher-Trapp, 2017), and the role of warm-rain in precipitation production (Plummer et al., 2018; Lasher-Trapp et al., 2018). However, the greater number of cases examined in this study compared to other COPE studies highlights the great diversity exhibited by clouds not only on different days but within a single day. This, in turn, illustrates the importance of observing not only cloud structure itself, but the environment in which the clouds are growing. Further, the observations suggest that while in some clouds, one process may appear to dominate ice formation, in another cloud—its near neighbor in some cases—an entirely different process may be important. This too underscores the importance of considering multiple processes and obtaining a diverse set of observations from many clouds in order to elucidate the importance of critical processes.



**Acknowledgements**

This work was funded by the U.S. National Science Foundation under grants AGS-1230292 and AGS-1230203 for the U.S. investigators, with UWKA participation funded by grant AGS-1441831. The work was also partly funded by the U.K. Natural Environment Research Council under grant NE/J023507/1. We acknowledge the

Centre for Environmental Data Analysis for storing archived COPE data and NCAS Atmospheric Measurement Facility for use of the NXPol. Finally, we would like to thank the hard work and dedication of the crew of the UWKA.

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



Table 1. Environmental and cloud conditions from the four days sampled in this study. The definition of what constitutes a penetration and updraft core is provided in the text.

| Case | CAPE [J kg$^{-1}$] | Cloud base $T$ [°C] | Cloud top $T$ [°C] | Cloud base to top wind shear | Cloud Droplet Conc. | # UWKA Pens | # Updrafts | Median max vertical wind [m s$^{-1}$] | Range of max vertical wind (25/75 quartiles) |
|---|---|---|---|---|---|---|---|---|---|
| 28 July | 136 | 9 | -13 | 1.2 x 10$^{-4}$ s$^{-1}$ | 375 cm$^{-3}$ | 47 | 17 | 5.7 | 3.5 to 9.6 m s$^{-1}$ (4.7 to 6.8 m s$^{-1}$) |
| 29 July | 301 | 11 | -25 | 1.0 x 10$^{-3}$ s$^{-1}$ | 300 cm$^{-3}$ | 63 | 20 | 10.2 | 3.1 to 14.9 m s$^{-1}$ (7.2 to 11.2 m s$^{-1}$) |
| 02 August | 615 | 12 | -20 | 5.2 x 10$^{-3}$ s$^{-1}$ | 600 cm$^{-3}$ | 66 | 34 | 8.2 | 3.0 to 18.2 m s$^{-1}$ (5.1 to 12.5 m s$^{-1}$) |
| 03 August | 247 | 11 | -16 | 1.7 x 10$^{-3}$ s$^{-1}$ | 325 cm$^{-3}$ | 49 | 13 | 7.4 | 3.1 to 14.3 m s$^{-1}$ (4.7 to 9.4 m s$^{-1}$) |



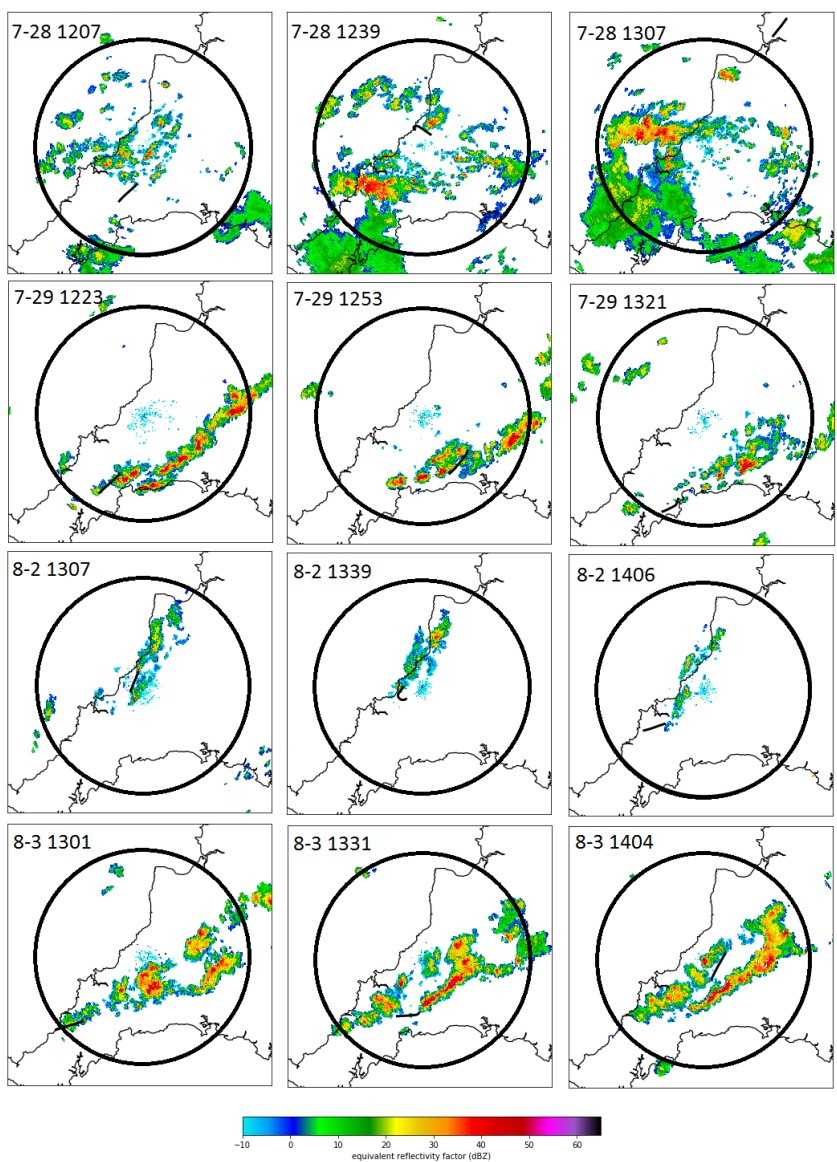

**Figure 1.** Reflectivity factor from the 3.5° elevation scan from NCAS radar for 3 times during a one-hour period on each of the study days. The first row shows scans from Jul 28, Jul 29 on the second, Aug 02 on the third, and Aug 03 on the fourth row. The times on each day correspond roughly to the time period centered on the flight of the UWKA. Range rings are shown for 50 km centered on the radar. The thick black line indicates the flight path of the UWKA for a 15-minute period centered on the time of the radar scan.



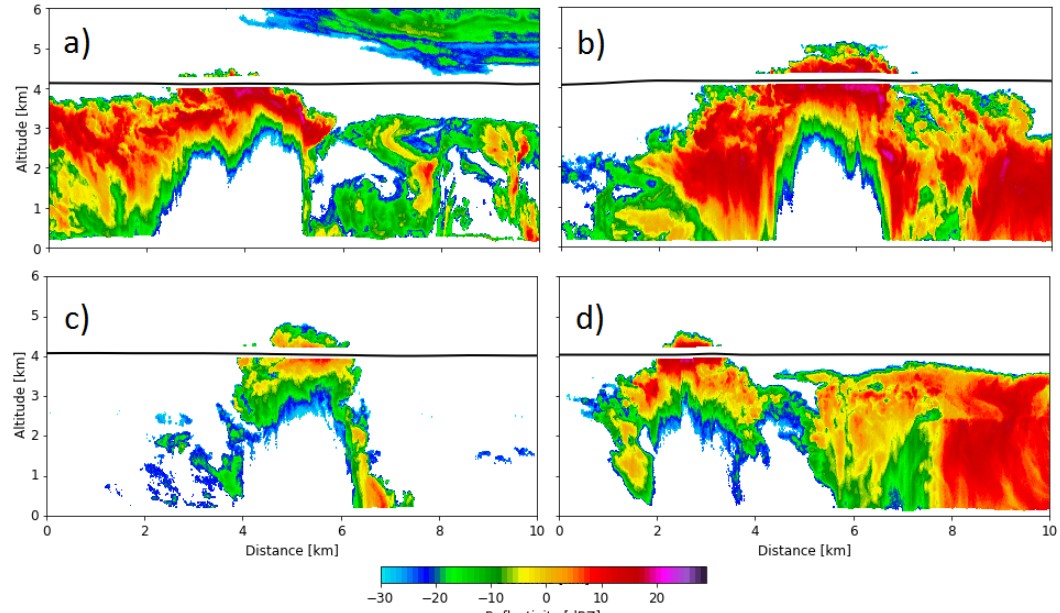

**Figure 2**. Example images or vertical profile of radar reflectivity from the WCR during penetrations on (a) July 28, (b) July 29, (c) Aug 02, and (d) Aug 03 at temperatures from -6 to -8 °C.



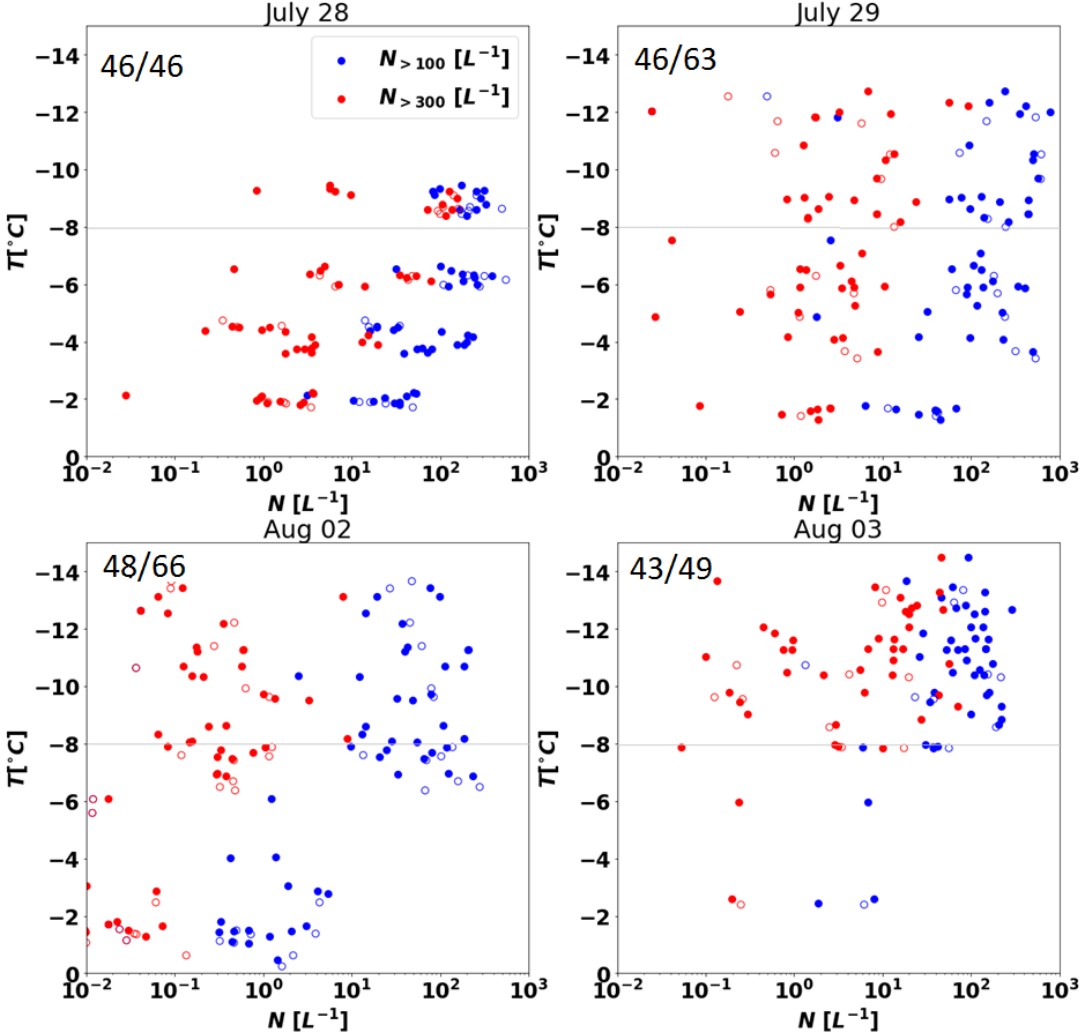

**Figure 3.** Median hydrometeor concentrations for all particles with D > 100 µm (blue) and D > 300 µm (red) at the corresponding temperature levels for all penetrations for the four days. Open circles represent those penetrations that meet the criteria for an updraft core (see text) and closed circles do not. The fractional number in the upper left of each graph represents the number of penetrations with median N > $10^{-2}$ L$^{-1}$. The denominator indicates the total number of penetrations on that day.




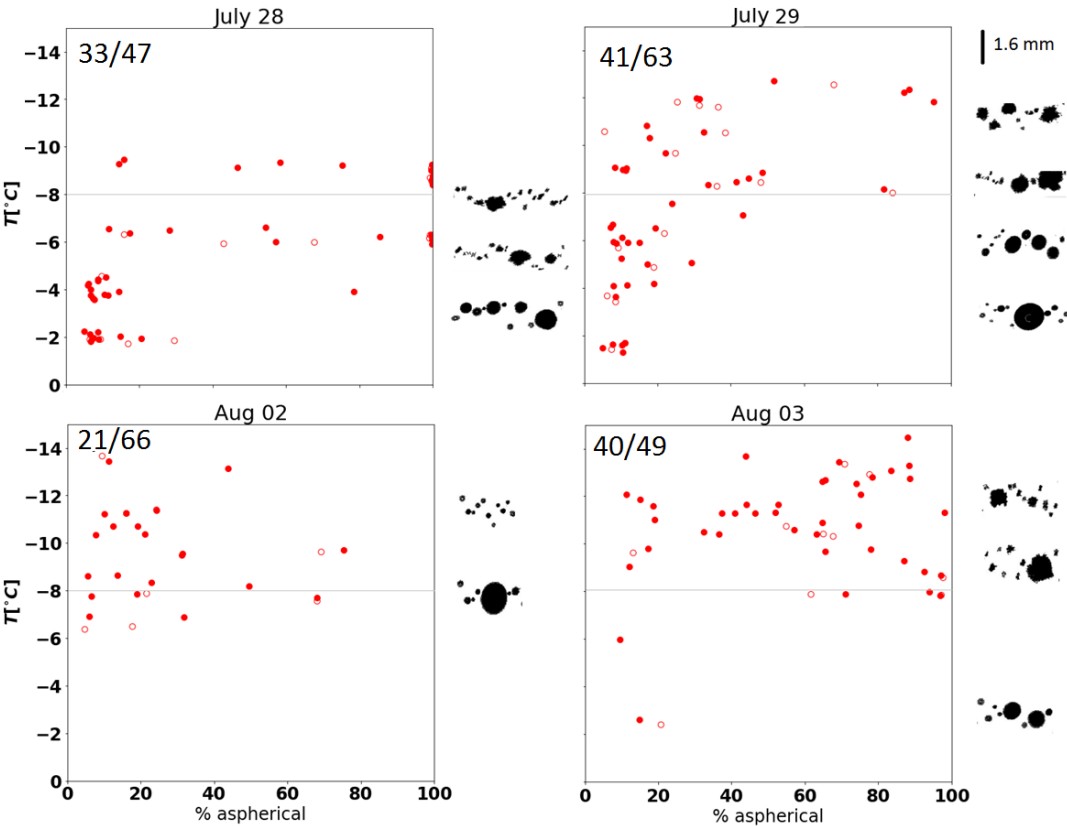

**Figure 4.** The percent of hydrometeors (in red) that were classified as non-spherical at the corresponding temperature for updraft cores (open circles) and penetrations not meeting updraft core criteria (dots) for the four days in the study. Only particles with D > 300 μm were considered. The fractional number represents the number of penetrations that contained at least 10 such particles. Example images from the OAP CIP of hydrometeors observed during penetrations at the indicated temperature are shown on the right side of the graph for each day. A scale for the images is shown in the upper left panel.





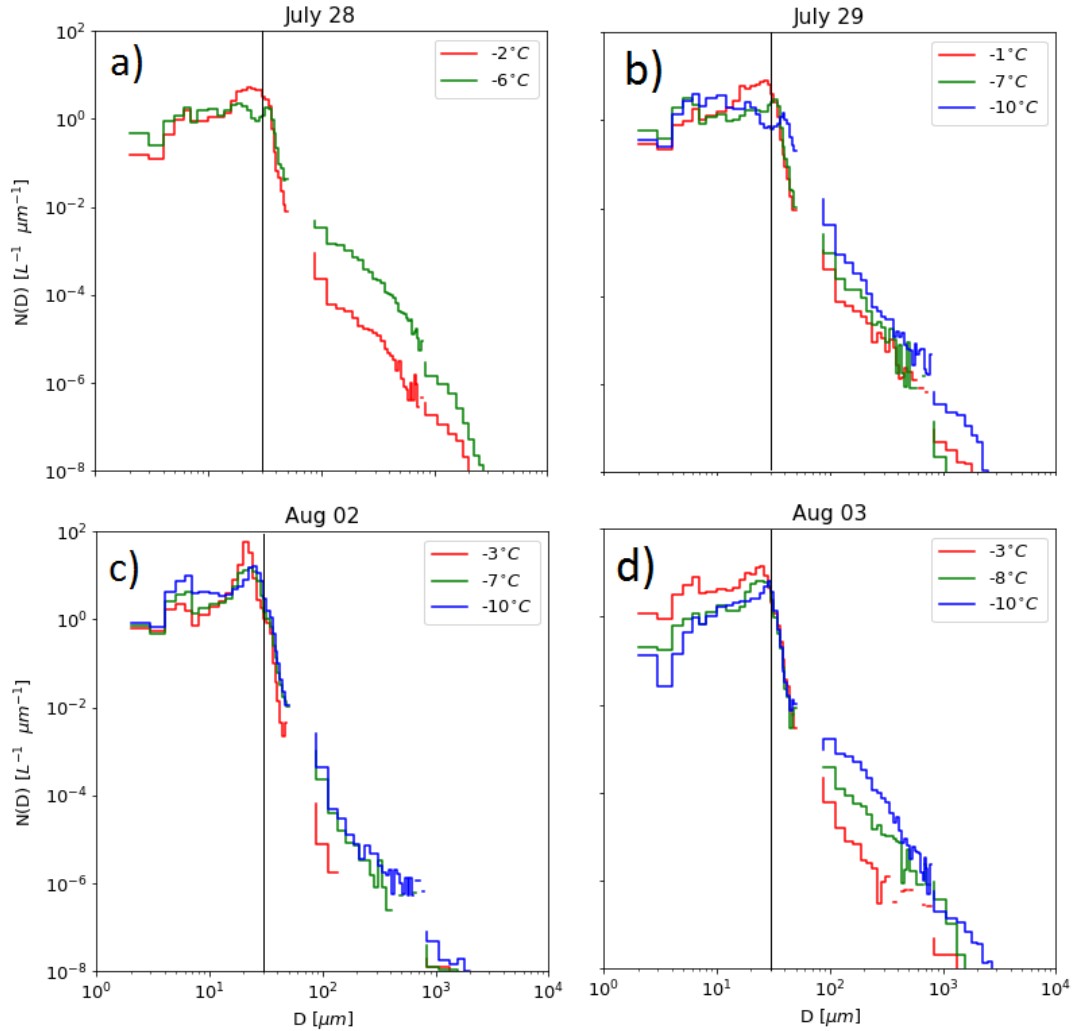

**Figure 5.** Mean *N(D)* from the CDP, CIP, and 2DP for the specified penetrations on (a) July 28, (b) July 29, (c) 02 August, and (d) 03 August. The solid black line denotes 30 μm, the minimum size water drops needed for the Hallett-Mossop process to occur.





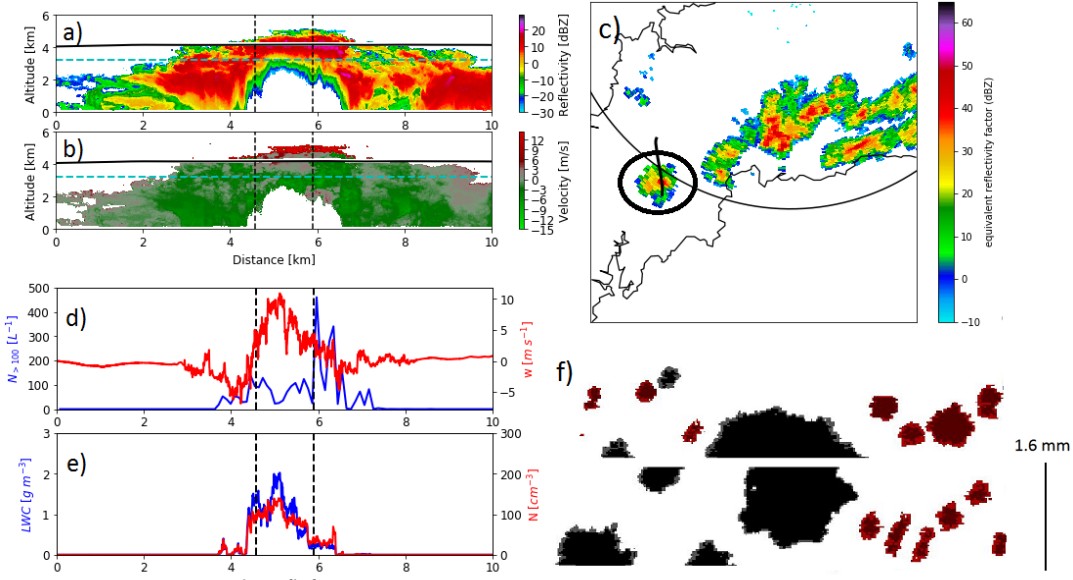

**Figure 6.** (a) WCR radar reflectivity and (b) vertical velocity for a penetration at -8℃ on 29 July with ice

phase precipitation. (c) PPI of NCAS radar reflectivity at 1.5 km MSL for the scan taken during the time

5    of the penetration. The thick black line indicates the UWKA flight track during the scan and the circle

indicates the clouds penetrated by the UWKA. (d) Time-series trace of hydrometeor concentration with

D>100 μm ($N_{>100}$; blue) and vertical wind ($w$; red) through the penetration. (e) Time-series trace of cloud

droplet concentration ($N$; blue) and cloud liquid water content ($LWC$; red) from the CDP. (f)

Representative hydrometeor images recorded by the CIP in the updraft core, with particles identified as

10    spherical colored blue, aspherical colored red, and particles not identified by the algorithm colored black.

Vertical dotted lines in (a,b,d,e) denote the boundaries of the updraft core.





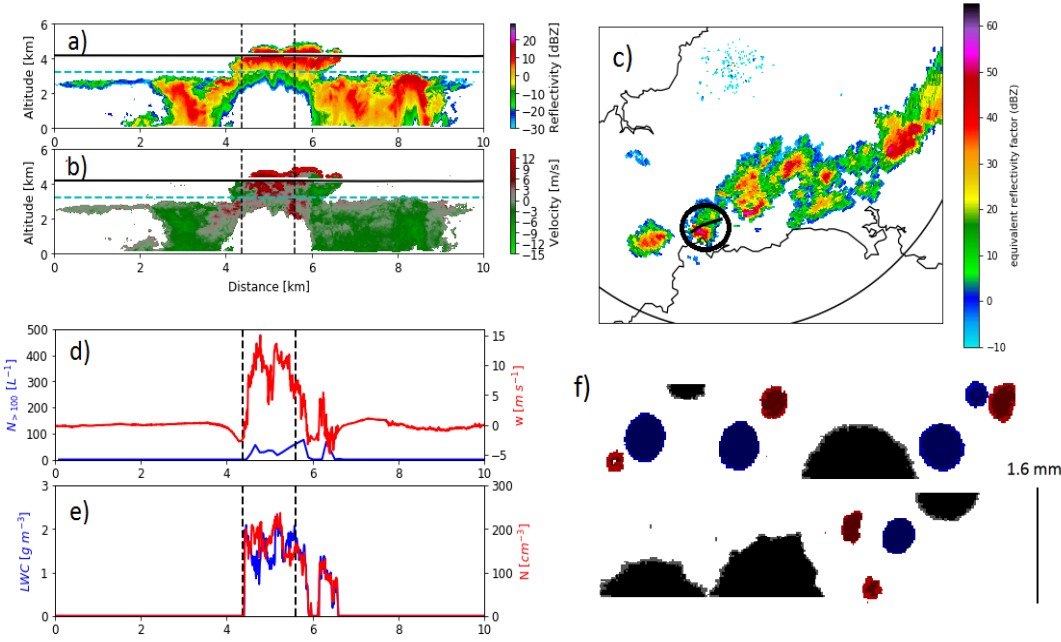

**Figure 7.** As Figure 6, but for the penetration at -8°C with mixed phase precipitation on 29 July.



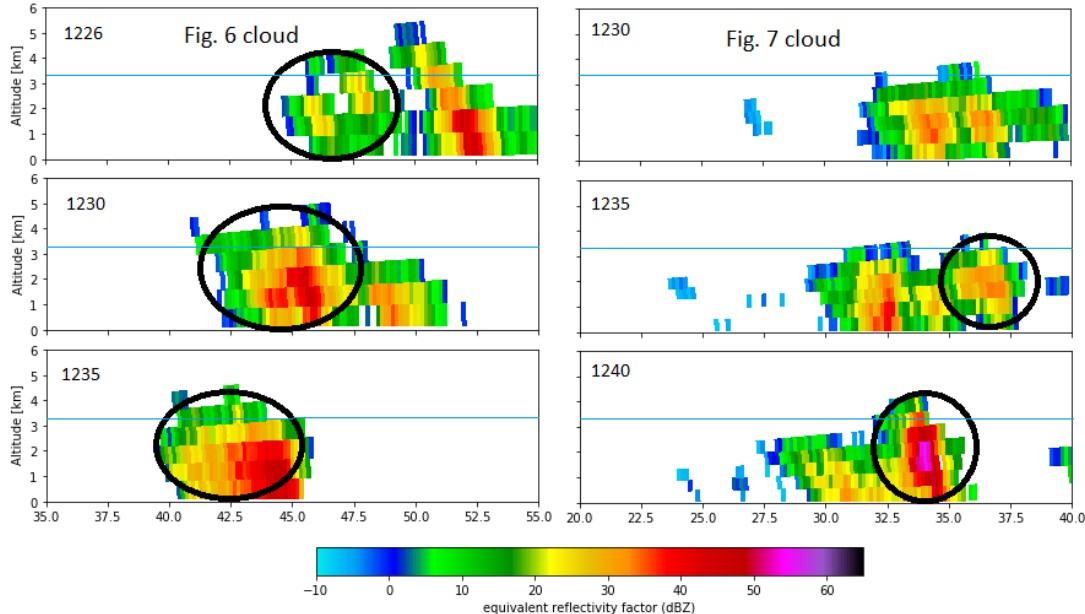

**Figure 8.** Timeseries of pseudo-range height indicator (RHI) plots of NCAS radar reflectivity on 29 July 2013 through the cloud in Fig. 6 (left column) and in Fig. 7 (right column). Circle indicates the location of the cloud penetrated by the UWKA. Blue line indicates location of 0°C isotherm.

