# Peer review of "Observations of the microphysical evolution of convective clouds in southwest United Kingdom"

_Atmospheric Chemistry and Physics, 2018_

## Referee Comment (RC1) · Anonymous Referee #1 · 14 May 2018

**Review of "*Observations of the microphysical evolution of convective clouds in southwest United Kingdom*"**

Four days' worth of microphysical and dynamical data area analyzed from the COnvective Precipitation Experiment (COPE) over Southwest England. The cases are chosen in a spectrum from one almost entirely glaciated by -10°C to one still primarily liquid at -13°C. The dynamic conditions also vary from low shear-low CAPE to high shear-high CAPE. Although the analysis and insight are not as deep as in the Taylor et al. COPE study, the work adds to a body of in-situ secondary ice production observations, and the discussion of ice recycling in different dynamic environments is interesting.

While the cases present a nice variety, I am wondering whether it would ease readability to refer to them with acronyms or names as opposed to dates. Table 1 is useful in this regard, but I did find myself flipping back and forth quite often to recall which case was which.

A strength of the work is that it considers different dynamic environments, but I wonder if this could not be made more quantitative. Could shear profiles / hodographs or CAPE evolution (The values in Table 1 are spatiotemporally averaged?) be included? This would make the discussion in Section 3a more rigorous. My other questions and comments are associated with specific lines or figures:

**Specific comments**

*Particle numbers in different size range*
   Page 5, Lines 10, 30-31; Page 6, Line 12 – Could you clarify where particle number concentrations in the different size ranges come from? "CDP sampled particles with diameter $2 < D < 50$ um"; "Concentrations of particles of $25 < D < 100$ um from the CIP are not reported in this study"; "Images …with diameters less than roughly 250 um are not included in the ice categorization". Where do the number concentrations between 50 and 125 um come from in Figure 5? Are the authors concerned that with a cutoff size of 250 um for ice that some fragments are not accounted for?

*Droplet size distribution broadening*
   Page 6, Lines 30-31; Page 10, Line 2 – I would make more explicit which droplet number concentrations were anomalously high and which were anomalously low on 02 August… for example stating on Page 6 that concentration of drops *with diameter greater than 30 um* "were orders of magnitude less." And on page 10 that there is a "larger cloud droplet number concentration" *of less than 30 um diameter.* Alternatively, the analysis concerning droplet SDs could be reworded in terms of the size distribution *broadening* or *narrowing* as in Lawson et al. 2017 *JAS*.

*Updraft dependence*
   Page 8, Lines 30-32 – I was rather surprised that neither the size-segregated hydrometeor concentrations nor the aspherical percentages had "any systematic difference … between observations obtained from cloud penetrations without updrafts." I think more discussion is warranted here because many studies have noted an influence of updraft to secondary ice production rates, e.g. Mossop 1976 *QJRMS*, Heymsfield and Willis 2014 *JAS*, Lawson et

al. 2015 *JAS* among others. And one might expect, given the highest percentage of strong updrafts on 02 August, that droplet shattering would be facilitated with a lofting of even larger droplets to high altitudes. That is not the case here, presumably because the aerosol loading is higher. But I do think this should be stately explicitly somewhere in the analysis.

Figure 5 – Is there a reason that the particle size distributions are not shown for the same temperatures for all cases? Clarifying which particle numbers are shown in which size ranges, the *N(D)* before the "break" in the spectra around 80 um, are exclusively droplet numbers, right? (If not, I am surprised that spectra at different temperatures overlap between 30 and 50 um.)

Section 3c – This is a nice comparison, but it would flow more logically to me if the fifth paragraph ("The two penetrations considered here…") came second (after "The observations demonstrate… few graupel particles"). This is the motivation of the comparison and the other paragraphs explain the differences.

Page 12, Lines 6-8 – This sentence is unwieldy. Could you say "*Variation in the spatiotemporal distribution of ice and precipitation production for these CAPE cases is likely due to a variety of ice production mechanisms.*"?

Figure 8 – This figure is not mentioned in the text. Some explanation should be incorporated, or it should be removed.

---

## Referee Comment (RC2) · Anonymous Referee #2 · 11 Jun 2018

The paper examines 4 convective cases from the recent COPE campaign using aircraft and radar data. I found the paper generally well written. The approach is largely qualitative and I would like to see more work done to provide quantification of secondary ice processes that would be useful to the modelling community. My comments are generally minor in that I do not expect them to undermine the main messages of the paper. However, I think they would strengthen the paper and make it more useful to the community.

General comment This was a multi aircraft campaign. The BAE146 data is referred to in terms of reports from the Taylor paper. Why wasn't the data included in the

analysis to confirm or extend the observations from the Wyoming King Air? I only mention this because whenever observationalists request resources for aircraft there are often cases made for the use of two aircraft. This would be an excellent opportunity to demonstrate the success of using multi-aircraft that future proposals can point to. Section 1 – there is also a recent COPE modelling study by Miltenberger (https://www.atmos-chem-phys.net/18/3119/2018/) and another by Yang that looks at updraughts from COPE that may provide some nice context (https://www.atmos-chem-phys.net/16/10159/2016/).

Specific comments: P7 line 17-27: I agree that threshold need to be chosen to make the analysis tractable but it would be worth a statement to say if there was any sensitivity to the choice of these thresholds (0.05g/m3, 300m, 100m, 1m/s, 3m/s) in terms of the results and conclusions drawn.

P8 line 30: this is a surprising result given the underlying hypothesis that secondary production is active and linked to processes in the updraft rather than the anvil regions. Is this lack of difference between the updraft and non-updraft region supported by the other aircraft observations? For the discussion - what do models show when segregated like this?

P9 line 1 – do you have plots of the penetration lengths as a function of T for the different days? Perhaps these figures would be improved by including some measure of the variability along the penetration. Could add 25th, 75th percentiles for example.

P9 line 1 – what do the droplet concentrations look like from CDP or something similar? Do they show a difference in and out of the updrafts?

P9 line 6 – are growing turrets the penetrations with updrafts?

P9 line 10– what was the strategy for sampling clouds with the UWKA? Was it the same on all days? Was the sampling strategy for the BAe146 the same? The difference in results suggests that it would be good to combine datasets from both aircraft to provide

a fuller picture of the cloud characteristics.

P9 line 35 – figure 5 is from the updraft penetrations. Have you got the same plot for the non-updraft penetrations?

P10 line 2- agreed that there are twice as many droplets near cloud base and that would lead to smaller droplets for the same liquid water content – but it will only by 20% smaller. For 2 aug, the cloud base is also warmer suggesting that more liquid water would be available that could offset the effect of increased droplet numbers. . .

P10 line 1 – some mention of the 30micron threshold here, but not its importance to the Hallett-Mossop process as suggested in the caption to figure 5. There should be some more discussion about this here or earlier in the paper.

P10 line 12 – Fig 4 2d imagery suggests that july 29th when secondary production was thought to be less effective also has large rimed particles present. . .

P10 line 20. If invoking the H-M process then I think you also need to comment on the conditions that are felt necessary for it to be active (e.g. p10 line 1 comment). Beside the temperature range there are other parameters such as the range of liquid droplet sizes present and accretion rate that could also be explored to understand if conditions satisfy what was observed in the laboratory. Additionally, to be useful to modellers some estimates of the splintering rate as a function of temperature, accretion rate etc would be a useful step.

P10 line 19. To be pedantic, the role of primary nucleation has not been ruled out. There was no ice nucleation information available, but there needs to be some discussion about the fact that these concentration likely outstrip the primary production rates. Perhaps using DeMott 2010 and tying that to observed large aerosol in the boundary layer is a means to estimating a bound for the primary ice nucleated particles. I see that this discussion occurs in section 4 but it might be good to combine this discussion with the comments about primary ice concentrations.

P10 line 22. Why can't the ice be carried up from the H-M zone to the colder temperatures?

P10 line 22-24. I think this is speculation that should be moved to the discussion.

P11 line 22. Concentrations – it would be good to quote the spatial scale over which this is appropriate to help with comparison to models.

P12 line 14. Can you comment on the requirement for smaller droplets alongside millimetre size droplets to allow H-M to proceed?

P12 in cloud temperature measurements are difficult. It might be worth commenting on this for situations where there are strong updrafts and latent heating occurring.

Conclusions. Between 1) and 2), I think it would be good to add a statement that primary ice concentrations based on DeMott 2010 and aerosol (D>0.5micron) measurements (need to add this analysis into results) are much lower than observed ice concentrations and therefore it appears that a secondary ice production mechanism was active. Conclusion 3. As mentioned earlier, the H-M process has a set of conditions defined from laboratory work that is more extensive than just the temperature range. Please could you assess whether all of the tests are passed? This would strengthen the assertion and/or motivate further laboratory work. Conclusion 5. I do not necessarily agree with this – see my comment above about the contending effect of a lower cloudbase. I think you can speculate about the effect of droplet number in the discussion, but I don't think it can be a robust conclusion.

---

## Author Comment (AC1) · 15 Aug 2018

Four days' worth of microphysical and dynamical data area analyzed from the COnvective Precipitation Experiment (COPE) over Southwest England. The cases are chosen in a spectrum from one almost entirely glaciated by -10°C to one still primarily liquid at -13°C. The dynamic conditions also vary from low shear-low CAPE to high shear-high CAPE. Although the analysis and insight are not as deep as in the Taylor et al. COPE study, the work adds to a body of in-situ secondary ice production observations, and the discussion of ice recycling in different dynamic environments is interesting.

While the cases present a nice variety, I am wondering whether it would ease readability to refer to them with acronyms or names as opposed to dates. Table 1 is useful in this regard, but I did find myself flipping back and forth quite often to recall which case was which.
A strength of the work is that it considers different dynamic environments, but I wonder if this could not be made more quantitative. Could shear profiles / hodographs or CAPE evolution (The values in Table 1 are spatiotemporally averaged?) be included? This would make the discussion in Section 3a more rigorous. My other questions and comments are associated with specific lines or figures:

**We thank the reviewer for their constructive comments. We share the reviewer's concern that the presentation of the four cases can be quite overwhelming especially given the wide variety of information. We tried different methodologies and acronyms, but all seemed to be rather confusing. The most logical way we found to label these cases for the reader would be to classify them as Cases A to D. In the revised manuscript, we refer to these alpha-numeric codings (introduced at the beginning of section 3) throughout sections 3 and 4. However, in the conclusions section we present the cases by their date.**

**Regarding the reviewer's comment on quantification of shear profiles and/or CAPE evolution, we are unable to provide an evolution of CAPE for the following reasons below.**

[Figure]

**Figure R1. Skew-T diagram of the sounding launch at 15 UTC 03 August. The black line denotes temperature, the red dashed line is dewpoint. The pink dashed line shows the temperature of a parcel lifted from the surface.**

[Figure]

**Figure R2. As Figure R1, but after the surface temperature is cooled to 16 degrees Celsius.**

**When calculating CAPE we carefully selected soundings that were closest to the aircraft penetrations in time. We also found that when we calculated CAPE there is known to be a large sensitivity to both the surface temperature used as well as due to the lifting due to the convergence line (Browning et al. 2007). We had to carefully check whether the calculated thermodynamic profiles were representative of the observed cloud bases and tops on those days. Figure R1 and R2 show an example of this sensitivity. Therefore, we decided to calculate CAPE based on the observed cloud bases and tops throughout the day. Using this process is impossible without detailed information about how cloud tops vary throughout the day to help verify the validity of the CAPE calculation. Therefore, we are unable to provide an evolution of CAPE throughout the day.**

[Figure]

**Figure R3. Hodographs for the 4 cases.**

The hodographs shown for the 4 cases in Figure R3 does provide a qualitative visual confirmation of calculations of shear provided in Section 3a. However, no real additional information is gained. We see unidirectional shear on all of the days, and the shear is the greatest on 02 August (Case C), the least on 28 July (Case A). Furthermore, we feel that the most representative calculation of shear for these cases is the shear from cloud base to cloud top, as that is what will impact the cloud dynamics and in turn the microphysics. Therefore, we feel that we have, to the best of our ability, provided the most quantitative assessment of the thermodynamic and dynamic profiles on the 4 days that we are able to provide based on the data set available.

**Specific comments**
*Particle numbers in different size range*
Page 5, Lines 10, 30-31; Page 6, Line 12 – Could you clarify where particle number concentrations in the different size ranges come from? "CDP sampled particles with diameter 2 < D < 50 um"; "Concentrations of particles of 25 < D < 100 um from the CIP are not reported in this study"; "Images …with diameters less than roughly 250 um are not included in the ice categorization". Where do the number concentrations between 50 and 125 um come from in Figure 5? Are the authors concerned that with a cutoff size of 250 um for ice that some fragments are not accounted for?

To clarify – the CDP measures particles with 2 < D < 50 um, the CIP nominally measures particles with 25 < D < 1600 um, and the 2DP nominally measures particles with 200 < D < 6.4 mm. In our processing of the data, since the concentrations from the CIP are unreliable at D < 100 μm, we removed any particles in the CIP data that had D < 100 um. For regions of overlap between the CIP and 2DP, we used CIP measurements for D<800 um and 2DP measurements for D>800 um. The 'gap' in the spectra in Figure 5, accounts for particles with 50 < D < 100 um. We have added a clarifying clause in the last paragraph of page 5 to clarify to the reader what size ranges are associated with which probes:

"To account for regions of diameter overlap between probes, and to remove significant uncertainty associated with poorly resolved particles from OAPs, for the remainder of the manuscript, concentrations of particles with $2 < D < 50$ μm are reported from the CDP, $100 < D < 800$ μm from the CIP, and $800$ μm $< D < 6400$ μm from the 2DP."

**Yes, since we do not factor in particles of less than 100 square pixels in our habit analysis, we are concerned that some of the smallest ice fragments are not accounted for. However, we cannot provide a reliable estimate of how many small ice fragments are being excluded from the analysis, as it is nearly impossible to reliably determine the shape of particles from the CIP that are less than 100 square pixels in area. However, any ice fragments that are produced are expected to rapidly grow in regions of ice supersaturation and significant cloud liquid water. Regardless, we acknowledge being unable to estimate the number of smaller crystals in our analysis with the addition of the following text in the last paragraph of section 2b:**

"...corresponds to hydrometeors with diameters less than roughly 250 μm. While this threshold excludes some small ice fragments, it is impossible to provide a reliable estimate of how many fragments are excluded."

*Droplet size distribution broadening*
Page 6, Lines 30-31; Page 10, Line 2 – I would make more explicit which droplet number concentrations were anomalously high and which were anomalously low on 02 August… for example stating on Page 6 that concentration of drops *with diameter greater than 30 um* "were orders of magnitude less." And on page 10 that there is a "larger cloud droplet number concentration" *of less than 30 um diameter.* Alternatively, the analysis concerning droplet SDs could be reworded in terms of the size distribution *broadening* or *narrowing* as in Lawson et al. 2017 *JAS*.

**On page 6 (second paragraph in Section 3), the discussion is focused on *precipitating drops* (i.e D > ~500 um). We clarify this in the revised manuscript stating:**
"…concentrations of drops with $D \sim 500$ um and greater were orders of magnitude less during Case C than on the other three days."

**On page 10 (second to last paragraph in section 3b, beginning with "Figure 5 shows…"), at the beginning the focus of the discussion is on *cloud droplets.* Here we use a threshold of 30 um, because droplets larger than this will likely begin to grow more effectively through collision-coalescence. The following sentence connects that a higher concentration of these drops will result in a smaller median diameter and narrower droplet spectrum:**
"…there is a larger concentration of cloud droplets with $D < 30$ μm in Case C. This is likely a due to the larger cloud droplet number concentration observed at cloud base in Case C (Table 1) and is consistent with a slower collision-coalescence process as expected given the smaller median droplet diameter, narrower droplet spectra, …"

*Updraft dependence*
Page 8, Lines 30-32 – I was rather surprised that neither the size-segregated hydrometeor concentrations nor the aspherical percentages had "any systematic difference … between observations obtained from cloud penetrations without updrafts." I think more discussion is warranted here because many studies have noted an influence of updraft to secondary ice production rates, e.g. Mossop 1976 *QJRMS*, Heymsfield and Willis 2014 *JAS*, Lawson et al. 2015 *JAS* among others. And one might expect, given the highest percentage of strong updrafts on 02 August, that droplet shattering would be facilitated with a lofting of even larger droplets to high altitudes.

That is not the case here, presumably because the aerosol loading is higher. But I do think this should be stately explicitly somewhere in the analysis.

**Quite honestly, we too were rather surprised by this finding. Our original analysis focused on just updraft regions. As we expanded our analyses to include cloud penetrations with 'no updraft' (actually updraft less than a 1 m/s threshold), we found no significant difference in our results. After much discussion, we attribute this to our sampling strategy. Since we targeted turrets as they first ascended to and just above the level of the aircraft, every penetration was a 'fresh turret'. All had hard, well-defined edges and none of the penetrations included anvil regions or clouds in their decaying stage. Because these turrets often extended above their equilibrium level, one may expect a rapid transition between a turret with an updraft and one whose updraft had weakened signficantly over just a few minutes. It appears the microphysical characteristics of these two types of turrets were quite similar. This discussion is added to the paragraph in section 3b in the revised manuscript.**

Figure 5 – Is there a reason that the particle size distributions are not shown for the same temperatures for all cases? Clarifying which particle numbers are shown in which size ranges, the *N(D)* before the "break" in the spectra around 80 um, are exclusively droplet numbers, right? (If not, I am surprised that spectra at different temperatures overlap between 30 and 50 um.)

**We showed size distributions at slightly different temperatures in Figure 5 because there was not always a penetration at the same temperature for each flight that was representative of the overall statistical analysis. We do not show concentrations of particles in the "break" (100 um in the revised manuscript as per the discussion earlier) as we do not have reliable measurements of the concentrations of particles between 50 and 100 um in diameter.**

Section 3c – This is a nice comparison, but it would flow more logically to me if the fifth paragraph ("The two penetrations considered here…") came second (after "The observations demonstrate… few graupel particles"). This is the motivation of the comparison and the other paragraphs explain the differences.

**We have rearranged section 3c according to this suggestion.**

Page 12, Lines 6-8 – This sentence is unwieldy. Could you say "*Variation in the spatiotemporal distribution of ice and precipitation production for these CAPE cases is likely due to a variety of ice production mechanisms.*"?

**We have replaced the sentence with your suggestion, except that we used "COPE" in place of "CAPE."**

Figure 8 – This figure is not mentioned in the text. Some explanation should be incorporated, or it should be removed.

**Figure 8 is referred to in the text in line 14 of page 11.**

**References:**

Browning, K.A., A.M. Blyth, P.A. Clark, U. Corsmeier, C.J. Morcrette, J.L. Agnew, S.P. Ballard, D. Bamber, C. Barthlott, L.J. Bennett, K.M. Beswick, M. Bitter, K.E. Bozier, B.J. Brooks, C.G. Collier, F. Davies, B. Deny, M.A. Dixon, T. Feuerle, R.M. Forbes, C. Gaffard, M.D. Gray, R. Hankers, T.J. Hewison, N. Kalthoff, S. Khodayar, M. Kohler, C. Kottmeier, S. Kraut, M. Kunz, D.N. Ladd, H.W. Lean, J. Lenfant, Z. Li, J. Marsham, J. McGregor, S.D. Mobbs, J. Nicol, E. Norton, D.J. Parker, F. Perry, M. Ramatschi, H.M. Ricketts, N.M. Roberts, A. Russell, H. Schulz, E.C. Slack, G. Vaughan, J. Waight, D.P. Wareing, R.J. Watson, A.R. Webb, and A. Wieser, 2007: The Convective Storm Initiation Project. *Bull. Amer. Meteor. Soc.,* **88**, 1939–1956, https://doi.org/10.1175/BAMS-88-12-1939

---

## Author Comment (AC2) · 15 Aug 2018

The paper examines 4 convective cases from the recent COPE campaign using aircraft and radar data. I found the paper generally well written. The approach is largely qualitative and I would like to see more work done to provide quantification of secondary ice processes that would be useful to the modelling community. My comments are generally minor in that I do not expect them to undermine the main messages of the paper. However, I think they would strengthen the paper and make it more useful to the community.

General comment This was a multi aircraft campaign. The BAE146 data is referred to in terms of reports from the Taylor paper. Why wasn't the data included in the C1 analysis to confirm or extend the observations from the Wyoming King Air? I only mention this because whenever observationalists request resources for aircraft there are often cases made for the use of two aircraft. This would be an excellent opportunity to demonstrate the success of using multi-aircraft that future proposals can point to.

**We thank the reviewer for their constructive comments.**

**COPE was a multi-aircraft campaign, with the intention of coordinating sampling strategy between platforms to leverage the strengths of each platform and provide much greater spatial and temporal coverage within the study domain. However, it is not possible to utilize the data from the BAe146 for this study for three important reasons. This study utilizes a statistical approach to characterize the microphysics near the top of turrets as they grew through specific levels. For this, it is important to be able to measure the distance to cloud top from the penetration level. In the UWKA data set this is provided by the WCR. Indeed it was shown in this study that more than 80% of cloud penetrations were within 1 km of cloud top. No such verification is available in the BAe146 data.**

**Because of the instrument complement on both aircraft, the UWKA flew at a higher altitude than the BAe146. During multi-aircraft missions, while the UWKA was penetrating the tops of turrets, the BAe146 was obtaining measurements much lower down in cloud. We feel that combining these two datasets would be mixing apples with oranges.**

**As detailed in Taylor et al. (2016b), the study utilizing -146 measurements from 3 August detailed how that aircraft made repeated penetrations in clouds along the convergence line in order to study the development along the line and also the time evolution. However this study focuses only on cloud characteristics as clouds first ascended through a specific level. These are two fundamentally different strategies.**

**Lastly, of the four cases examined in this study, the -146 only flew on 29 July and 03 August. No additional data would be available for 02 August or 28 July. For these three reasons we believe it best not to include the -146 data in this study but rather use this study to provide a compliment to the results of Taylor et al. (2016b).**

**In the revised manuscript, we add additional text at the beginning of section 3b describing in more detail the sampling strategy of the UWKA for this study.**

**We have embedded the responses to the reviewer's specific comments in bold.**

Specific comments:

Section 1 – there is also a recent COPE modelling study by Miltenberger (https://www.atmos-chem-phys.net/18/3119/2018/) and another by Yang that looks at updraughts from COPE that may provide some nice context (https://www.atmos-chemphys.net/16/10159/2016/).

**We have added text in Section 1 briefly describing the key points of these two papers as they relate to the current study in order to provide a better overview of previous COPE studies relevant to current work.**

P7 line 17-27: I agree that threshold need to be chosen to make the analysis tractable but it would be worth a statement to say if there was any sensitivity to the choice of these thresholds (0.05g/m3, 300m, 100m, 1m/s, 3m/s) in terms of the results and conclusions drawn.

**We conducted various sensitivity tests to the choice of the thresholds for LWC, updraft width, and the minimum velocity. We found that our overall conclusions were not sensitive to the choice of our thresholds. We have attached an example of a sensitivity test where we define a penetration using a 0.03 g/m3 threshold and one with a 0.05 g/m3 in Figures R1 and R2 that show the analyses in Figures 3 and 4 using these thresholds. In Figure R3 we have attached a version of Figures 3 and 4 using a minimum penetration (or updraft) depth of 300 m in place of 100 m. We notice little change in the conclusions drawn when such analyses are conducted with these new thresholds. In order to summarize these sensitivity tests, we added a sentence to the first paragraph in Section 3b:**

*"Sensitivity tests conducting analyses using differing thresholds for LWC and updraft width showed that this conclusion was insensitive to the thresholds used to define a penetration or updraft core (not shown)."*

[Figure]

**Figure R1. (left) The median total number concentration and (right) percentage of aspherical particles in each penetration defined by *LWC* > 0.03 g m⁻³.**

[Figure]

**Figure R2. As Figure R1, but defining a penetration using a LWC > 0.05 g m⁻³ threshold.**

[Figure]

**Figure R3. As Figure R1, but with a minimum penetration (updraft) length of 300 m.**

P8 line 30: this is a surprising result given the underlying hypothesis that secondary production is active and linked to processes in the updraft rather than the anvil regions. Is this lack of difference between

the updraft and non-updraft region supported by the other aircraft observations? For the discussion - what do models show when segregated like this?

**Reviewer 1 had a similar comment, below is our response to reviewer 1. It should also be noted that our sampling strategy focused on turrets as they were just passing through the UWKA level, so those devoid of updrafts were not anvil regions, but rather likely turrets that had just began transitioning.**

**Our original analysis focussed on just updraft regions. As we expanded our analyses to include cloud penetrations with 'no updraft' (actually updraft less than a 1 m/s threshold), we found no signficant difference in our results. After much discussion, we attribute this to our sampling strategy. Since we targeted turrets as they first ascended to and just above the level of the aircraft, every penetration was a 'fresh turret'. All had hard, well-defined edges and none of the penetrations included anvil regions or clouds in their decaying stage. Because these turrets often extended above their equilibrium level, one may expect a rapid transition between a turret with an updraft and one whose updraft had weakened signficantly over just a few minutes. It appears the microphysical characteristics of these two types of turrets were quite similar. This discussion is added to the paragraph in section 3b in the revised manuscript.**

P9 line 1 – do you have plots of the penetration lengths as a function of T for the different days? Perhaps these figures would be improved by including some measure of the variability along the penetration. Could add 25th, 75th percentiles for example.

**Figure R4 shows the length of each penetration. The penetrations lengths generally varied between 0.2 km to 2 km, and wording has been added to this paragraph to state the overall lengths of each penetration:**

"The penetrations ranged from 0.2 km to 2 km in length."

[Figure]

**Figure R4. The length of the penetration (solid circles) or updrafts (hollow circles) as a function of temperature.**

**Regarding the inclusion of quartiles of these penetration plots; An early version of this manuscript had included just such information, however, we found that it was difficult to create a figure that was easily readable with this information included. Further, it did not alter the outcome of the analysis and therefore, in the end, we decided to only include the medians.**

P9 line 1 – what do the droplet concentrations look like from CDP or something similar? Do they show a difference in and out of the updrafts?

**The range of CDP concentrations for all of the penetrations: 98 cm$^{-3}$ on July 28, 75 cm$^{-3}$ on July 29, 175 cm$^{-3}$ on Aug 02 and 96 cm$^{-3}$ on Aug 03. We have added these values to Table 1 and modified the wording to say that there is no systematic difference between the ice concentrations and habits between updrafts and penetrations.**

P9 line 6 – are growing turrets the penetrations with updrafts?

**We attempted to sample *only* growing turrets. Turrets that were obviously collapsing and/or did not have sharp, well-defined edges were not sampled. We expect those that were still actively rising at the time of penetration were the ones that contained updrafts. Those that did not contain updrafts had likely lost all of their buoyancy at the time of penetration.**

P9 line 10– what was the strategy for sampling clouds with the UWKA? Was it the same on all days? Was the sampling strategy for the BAe146 the same? The difference in results suggests that it would be good to combine datasets from both aircraft to provide C2 a fuller picture of the cloud characteristics.

**We provide the answer to this in our first response above. Yes, we did seek to sample the clouds with the same strategy on all four days and it did differ from the strategy employed by the -146. The first paragraph of section 3b in the revised manuscript contains discussion of the sampling strategy.**

P9 line 35 – figure 5 is from the updraft penetrations. Have you got the same plot for the non-updraft penetrations?

[Figure]

**Figure R5. As Fig. 5 in the manuscript, but for penetrations without updrafts.**

**Fig R5 is the same as Figure 5 in the manuscript except for the penetrations without updrafts. For the precipitation size particles, we see similar trends compared to the analysis in the manuscript, except with fewer precipitation particles on 29 July outside of the updrafts. However, for the cloud droplets, we do generally see reduced number concentrations all days except July 29 as we approach colder temperatures, as presumably without an updraft the precipitation particles would act to deplete the cloud droplets via accretion and secondary ice production.**

P10 line 2- agreed that there are twice as many droplets near cloud base and that would lead to smaller droplets for the same liquid water content – but it will only by 20% smaller. For 2 aug, the cloud base is also warmer suggesting that more liquid water would be available that could offset the effect of increased droplet numbers. . .

**The cloud bases between the days did not differ by more than 4 degrees Celsius. The modelling of the 02 August case by Lasher-Trapp et al. (2018) is consistent with the notion that the increased droplet numbers provided narrower droplet size distributions that led to the inhibition of the warm rain process.**

P10 line 1 – some mention of the 30 micron threshold here, but not its importance to the Hallett-Mossop process as suggested in the caption to figure 5. There should be some more discussion about this here or earlier in the paper.

**The Hallet-Mossop threshold is 24 μm. In Figure 5, this vertical line is meant to provide a reference diameter for the reader to easily compare distribution modes between the 4 days. We do not make an assertion that there are more (or fewer) droplets that have achieved this threshold diameter and**

therefore act as the principal controlling factor for the onset of H-M. Rather we assert that the lack of warm-rain production on 02 Aug leads to a dearth of drops that can freeze and become 'instant rimers'. In the revised manuscript we remove the reference to 'minimum size water drops needed for the Hallett-Mossop process to occur.' from the caption in Figure 5.

P10 line 12 – Fig 4 2d imagery suggests that july 29th when secondary production was thought to be less effective also has large rimed particles present. . .

**The 2D images in Figure 4 are *examples* of particles from the penetrations, but not necessarily every penetration has particles with those habits. The aspherical percentage on July 29 at temperatures from -3 to -8 C (and even lower temperatures) is significantly less than on July 28 and August 03, suggesting that ice production at those levels (ie secondary production) is less effective on that day.**

P10 line 20. If invoking the H-M process then I think you also need to comment on the conditions that are felt necessary for it to be active (e.g. p10 line 1 comment). Beside the temperature range there are other parameters such as the range of liquid droplet sizes present and accretion rate that could also be explored to understand if conditions satisfy what was observed in the laboratory. Additionally, to be useful to modellers some estimates of the splintering rate as a function of temperature, accretion rate etc would be a useful step.

**We agree with the reviewer that a more quantitative analysis of the splintering production rate and accretion rate of the Hallett-Mossop process would strengthen the paper. However, there are two quantities required by the calculation of the measured and predicted splinter production rate from Harris-Hobbs and Cooper (1987 that are difficult to quantify with our dataset. One is the number concentration of columns of sizes 87 to 140 µm that are needed to calculate the measured splintering rates. The other is the number concentration of graupel particles that is required to calculate the predicted splintering rates according to the laboratory studies. There are large uncertainties in the measurements taken from CIP and 2DP probes for both such measurements, so therefore it is important to check the sensitivity of such process rate calculations to the measured concentrations of columns and graupel.**

[Figure]

**Figure R6. The splinter production rate measured versus those rates predicted using the methodology from Harris-Hobbs and Cooper (1987) assuming that all particles with D of 100 to 140 μm are columns (left) and all particles with D > 800 μm are graupel. (middle) as (left) but assuming only half of the particles with D > 800 μm are graupel. (right) as (left) but assuming that only half of the particles with D from with D of 87 to 140 μm are columns. Only penetrations with aspherical percentages > 80 percent are included to ensure that most of the precipitation is ice.**

**Figure R6 shows the calculated measured process rates compared against predicted process rates in the penetrations. It shows that the conclusions drawn can be very sensitive to the number concentration of both columns and graupel. For example, in the middle plot one can see that, for many of the penetrations on July 28, the process rates are within an order of magnitude of each other but this ceases to be the case for the other two plots. Therefore, with this level of uncertainty, it is nearly impossible to quantify whether the observed splinter production rates agreed with those predicted from the laboratory studies and we therefore chose not to include these in the paper.**

P10 line 19. To be pedantic, the role of primary nucleation has not been ruled out. There was no ice nucleation information available, but there needs to be some discussion about the fact that these concentration likely outstrip the primary production rates. Perhaps using DeMott 2010 and tying that to observed large aerosol in the boundary layer is a means to estimating a bound for the primary ice nucleated particles. I see that this discussion occurs in section 4 but it might be good to combine this discussion with the comments about primary ice concentrations.

**The original manuscript included text in the discussion demonstrating that the measured ice crystal concentrations were orders of magnitude higher than those that would be predicted by DeMott et al. (2010) {Paragraph 1, section 4 original manuscript}. In the revised manuscript we modified that to include a reference to the analysis from Taylor et al. (2016a)'s paper on the observed aerosol concentrations during COPE. They predicted that the concentrations of INP using the DeMott et al. (2010) parameterizations applied to measured boundary layer aerosol concentrations that the number of INP would range from 0.1 to 10 $L^{-1}$. Therefore, this shows that secondary ice production mechanisms must be occurring.**

References:

DeMott, P. J., Prenni, A.J., Liu X., Kreidenweis S. M., Petters M. D., Twohy C. H., Richardson M. S., Eidhammer, T., and Rogers, D.C.: Predicting global atmospheric ice nuclei distributions and their impacts on climate. Proc. Natl. Acad. Sci., 107, 11217–11222, doi:10.1073/pnas.0910818107, 2010

Harris-Hobbs, R. L., and W. A. Cooper: Field Evidence Supporting Quantitative Predictions of Secondary Ice Production Rates. J. Atmospheric Sci., 44, 1071–1082, doi:10.1175/1520-0469(1987)044<1071:FESQPO>2.0.CO;2, 1987

Lasher-Trapp, S., Kumar, S., Moser. D.H., Blyth, A.M., French. J.R., Jackson, R.C., and Plummer. D.M., On different microphysical pathways to convective rainfall, J. Appl. Meteor. Climatol., under review, 2018

Taylor, J. W., Choularton, T. W., Blyth, A. M., Flynn, M. J., Williams, P. I., Young, G., Bower, K. N., Crosier, J., Gallagher, M. W., Dorsey, J. R., Liu, Z., and Rosenberg, P. D.: Aerosol measurements during COPE:

composition, size, and sources of CCN and INPs at the interface between marine and terrestrial influences, Atmos. Chem. Phys., 16, 11687-11709, https://doi.org/10.5194/acp-16-11687-2016, 2016a.

Taylor, J. W., Choularton, T. W., Blyth, A. M., Liu, Z., Bower, K. N., Crosier, J., Gallagher, M. W., Williams, P. I., Dorsey, J. R., Flynn, M. J., Bennett, L. J., Huang, Y., French, J., Korolev, A., and Brown, P. R. A.: Observations of cloud microphysics and ice formation during COPE, Atmos. Chem. Phys., 16, 799-826, https://doi.org/10.5194/acp-16-799-2016, 2016b.

---

## Author Response (AR2)

I am happy with most of the responses to my comments and i am pleased to see that the authors have computed the splinter production rates.

The authors then go on to use the computations to indicate that they should not include these results in the paper. Here i disagree and think that including these results would be useful for the community for two reasons.

**We would like to thank the reviewer for their constructive review of the revised manuscript. We agree with most of the reviewer's points listed here and have incorporated them in the paper, but there are a couple of counterpoints that we would like to make. Detailed responses are embedded below.**

i) If the analysis suggests that it is not possible to estimate splinter rate from aircraft measurements with current technology, then this is a result that it would be good to share.

**We think that extending this result to all current technology is not applicable in this case as there are probes that provide better capabilities than was used during COPE. For example, the Cloud Particle Imager was not used during COPE, but that probe can provide high resolution photographs of precipitation particles and would have aided in determining the presence of graupel which the Harris-Hobbs and Cooper calculation depends upon.**

ii) Using their approach the authors could generate a diagram that shows the ratio of predicted to measured splinters as a function of fraction of particles assumed to be columns and the fraction assumed to be graupel?

The authors could talk about what would be required for the data to be in agreement with the laboratory results and whether there are techniques that could be used to constrain that fraction.

**Such a diagram has been added to the discussion of secondary ice production processes along with a paragraph discussing the results. In general, we see that, for agreement, we need to assume that greater than 50% of the particles imaged by the 2DP are graupel.**

In addition, here are some thoughts related to the current analysis:

Is there a need to worry about column fraction, given that the contribution to number concentration from primary nucleation is thought to be insignificant? The ratio of predicted to measured splinter production rate would be dependent upon graupel fraction only.

**Harris-Hobbs and Cooper (1987) assumed in their calculations that most of the ice originated from secondary ice production and showed evidence to support their assumption, so the reviewer raises a valid point. We no longer consider the percentage of particles identified as columns in our uncertainty analysis.**

From fig. R6 it looks like the graupel fraction would need to be ~0.1 of the >800 micron sized particles for the centroids of each case's data is on the 1:1 line? It might be possible to do an assessment using human particle shape interpretation on some representative imagery to estimate if the graupel fraction is 1 in 10 of the particles larger than 800 microns?

5 **We thank the reviewer for this helpful comment. We have added a manual analysis of the CIP imagery for the cases in Figure 6 and Figure 7. Using manual image identification of the CIP images for the case in Figure 6, all of the images with maximum dimensions larger than 800 microns appeared to be graupel, and for the case in Figure 7, 7 of the 9 images from the CIP were graupel, so this would provide us with a range of 75% to 100% as a range of maximum graupel for the two case studies.**

10 **Therefore, this shows that assuming that at least 50% of the images are graupel is not unreasonable and shows greater evidence of the Hallet-Mossop process, this has been added to the manuscript. We also demonstrate that more rigorous quantification would require a probe that can more readily identify graupel from raindrops for particles of $D > 800$ µm, as if we assume that fewer than 50% of the particles are graupel, an order of magnitude difference between the observed and theoretical**
15 **splintering rates is seen.**

Therefore, i would like to see the inclusion of a discussion that outlines the findings and limitations of their splinter rate analysis.

**We have added a figure and a paragraph in the paper discussing the findings and limitations of our splinter rate calculations.**

[revised manuscript text omitted]